# Molecular docking and MD simulations predicted quercetin as a potent human interleukin-1 beta (hIL1β) inhibitor for improved endodontic disease management

Nezar Boreak🆔*, Shroog Ali Almasoudi, Abdulelah Alharbi🆔, Mona Judayba, Shahd Tahrei, Atyaf Abu eishah, Taghreed Ahmed Madkhali, Mashael Ali Hattan, Maryam Hassan Majrashi, Huda Ali Daak, Amani Hakami

Department of Restorative Dental Sciences, College of Dentistry, Jazan University, Jazan, Saudi Arabia

* nboraak@jazanu.edu.sa

## Abstract

IL1β, a pro-inflammatory cytokine, is a key mediator in the inflammatory processes linked to endodontic disorders. Studies have shown that IL1β production is elevated in symptomatic periapical lesions, highlighting its involvement in inflammation and lesion development. Elevated levels of IL1β correlate with larger lesion sizes and increased inflammation in periapical tissues. Given its role in inflammation, IL1β represents a potential therapeutic target for endodontic diseases, including the use of IL1β inhibitors. The present study used molecular docking and MD simulations to identify small molecule inhibitors of hIL1β. A small library of 329 plant-derived natural compounds was screened against hIL1β, the top five hits were selected based on their binding affinity and score and docked with hIL1β. Molecular docking results showed that Quercetin has the highest binding affinity (−10.3 kcal/mol) and exibits favorable interactions with hIL1β compared to the other four hits. Based on these observations, Quercetin was further subjected to MD simulations with hIL1β. MD trajectories were used to determine the interaction and stability of the hIL1β-Quercetin complex using various parameters, such as RMSD, RMSF, Rg, SASA, and hydrogen bond count of the apo hIL1β and Quercetin-hIL1β complex. Consistent RMSD, RMSF, Rg, and SASA values indicated the stability of the complex. Furthermore, hydrogen bond count emphasized the Quercetin's role as a non-disruptive binder. Moreover, secondary structure analysis, PCA, and calculations of Gibbs free energy revealed minimal structural changes and highlighted the stable conformational state of hIL1β upon Quercetin binding, suggesting a stabilizing effect of Quercetin. These observations suggest Quercetin's potential in the development of new treatments for endodontic diseases, which may lead to improved clinical outcomes and reduced recurrence rates.

**Data availability statement:** Yes - all data are fully available without restriction; All relevant data are within the paper.

**Funding:** The authors appreciatively acknowledge the funding of the Deanship of Graduate Studies & Scientific Research, Jazan University, Saudi Arabia, over Project Number: GSSRD-24. The funders had no role in study design, data collection and analysis, decision to publish, or preparation of the manuscript.

## Introduction

Endodontic disease involves inflammation and infection of the tooth pulp. While recent studies have detected a variety of microorganisms within endodontic lesions, including fungi and viruses, bacterial infections continue to be recognized as the leading cause of these conditions [1]. Root canal treatment is a common procedure aimed at preserving teeth that have suffered from extensive decay or infection [2–4]. While this treatment has a high success rate of 86%−98%, post-treatment endodontic diseases (PTED) such as persistent, recurrent, or emerging apical periodontitis can occur in up to 15% of cases [5]. PTED is characterized as an "endodontic failure," necessitating further clinical intervention [6]. This condition arises due to the interplay between microbial infection and the host immune response, leading to significant inflammation and bone resorption around the affected tooth [7].

Inflammatory mediators, including interleukins, tumor necrosis factor, and matrix metalloproteinases, play crucial roles in the pathogenesis of PTED [6,7]. Among these, Interleukin-1β (IL1β) is a prominent proinflammatory cytokine that significantly contributes to the inflammatory cascade in PTED [5,8]. IL1β is primarily produced by macrophages and dendritic cells, although other cell types like gingival fibroblasts, periodontal ligament cells, and osteoblasts can also secrete this cytokine [9,10]. The production of IL1β is a two-step process involving its synthesis as an inactive precursor (pro-IL1β) followed by proteolytic cleavage by caspase-1 to generate the active form [9,11]. This process is tightly regulated by pattern recognition receptors (PRRs) such as Toll-like receptors (TLRs), which recognize pathogen-associated molecular patterns (PAMPs) and damage-associated molecular patterns (DAMPs) [12,13]. The activation of the NLRP3 inflammasome is a crucial step in the maturation of IL1β [14]. IL1β significantly contributes to the inflammatory milieu in endodontic disease [5]. It enhances local blood flow, recruits leukocytes, and promotes neutrophil infiltration at the inflammation site [15]. Furthermore, IL1β is a potent inducer of bone resorption [16]. It stimulates the production of matrix metalloproteinases, which are involved in the degradation of extracellular matrix components, thereby exacerbating tissue destruction and bone loss [17,18].

Despite the high success rate of root canal treatments, the occurrence of PTED poses a significant challenge in endodontics. The persistent inflammation and bone resorption associated with PTED highlight the need for improved diagnostic and therapeutic approaches [5]. IL1β, as a key mediator in the inflammatory response of PTED, presents a promising target for therapeutic intervention.

Natural small molecule inhibitors offer several advantages, including ease of synthesis, potential for oral administration, and cost-effectiveness compared to biologics [19,20]. By inhibiting IL1β activity using small molecules, it may be possible to reduce persistent inflammation and improve treatment outcomes in endodontic infections and their complications. Therefore, this study aims to identify and evaluate small molecule inhibitors of IL1β through in silico approaches as a step toward developing adjunctive therapeutics for managing endodontic and post-treatment endodontic diseases. Flavonoids are a class of polyphenolic compounds found abundantly in fruits, vegetables, and other plant sources [21]. They exhibit a wide range of biological

activities, including anti-inflammatory, antioxidant, and antimicrobial properties [22–26]. Previous studies have demonstrated the potential of flavonoids in modulating inflammatory pathways, suggesting their utility in treating inflammatory diseases [27,28]. However, the specific effects of flavonoids on IL1β-mediated inflammation in PTED remain underexplored. Current study identified Quercetin as a top hit with inhibitory action against hIL1β. While considerable progress has been made in understanding Quercetin's therapeutic potential, its precise impact on IL 1β-driven inflammation across multiple cell types highlights an important opportunity for further investigation. Prior studies have shown that Quercetin suppresses IL 1β-induced cytokines and chemokines in ARPE 19 retinal cells via MAPK and NF κB pathways [29], inhibits proliferation and MMP, COX 2, and PGE2 production by rheumatoid synovial fibroblasts [30], and attenuates inflammatory pain by reducing oxidative stress and cytokine release [31]. Additionally, it selectively downregulates IL6 secretion from human mast cells in response to IL1 stimulation [32]. These converging lines of evidence point to Quercetin as a potent modulator of IL 1β-mediated inflammation across diverse tissue types. Given the documented anti-inflammatory properties of flavonoids, exploring their impact on IL1β could uncover new therapeutic options. The rationale for this research lies in the need to identify effective treatments that can directly target the inflammatory processes central to endodontic diseases and PTED.

## Materials and methods

### Computer server and online bioinformatics resources

A High-end dual-booted workstation (Intel Xeon W-3400 CPU, processor with 32 cores) running Windows 10 and Ubuntu 2020 beta was used to carry the molecular docking and MD simulations. AutoDock (v1.2.3); AutoDock is one of the most cited tools for molecular docking [33]. It uses a scoring function to predict the binding affinity between a ligand and its target protein. The software combines empirical free energy force fields with a Lamarckian genetic algorithm to search for the best binding sites. GROMACS (GROningen MAchine for Chemical Simulations); GROMACS is a highly efficient software package for MD simulations [34]. It is used to simulate the Newtonian equations of motion for systems with hundreds to millions of particles, making it suitable for studying the time evolution of ligand-protein complexes. We used BIOVIA Discovery Studio Visualizer (v21.1.0.20298), San Diego: Dassault Systèmes, 2021and PyMOL [35] were used to visualise and to determine the interactions between ligand and protein complexes. SwissADME (http://www.swissadme.ch/) and pkCSM (https://biosig.lab.uq.edu.au/pkcsm/) online servers were used to determine the ADMET properties and toxicity associated with the ligand molecules [36,37].

### Retrieval and preparation of the target molecule

The three-dimensional structure of the human interleukin-1 beta (hIL1β) was obtained from the Protein Data Bank (PDB) (https://www.rcsb.org/), bearing PDB ID: 8C3U [38]. The structure has a resolution of 1.95 Å and R-Factor (Observed) as 0.2093. The structure was retrieved as a complex with an antagonist and was prepared for molecular docking with a standard receptor protocol using AutoDock Tools [39]. The preparation involved removing non-essential molecules, including water, ions, and any co-crystallized ligands not involved in the study. Hydrogen atoms were added to ensure correct protonation states, and bond orders were adjusted as needed. Subsequently, the protein structure was optimized through energy minimization, using an appropriate force field to relieve steric clashes and ensure stability. The final prepared structure was then saved in the appropriate PDBQT format for subsequent molecular docking studies.

### Retrieval and preparation of the ligand molecules

A small flavonoid library of 329 plant-derived natural compounds was selected from the NPACT Database (https://webs.iiitd.edu.in/raghava/npact/index.html) [40], with many of these small molecules gaining recognition over the past decade as potential therapeutic agents for various diseases [41–43]. The 2D structures of the ligands were obtained in.mol format

and converted into 3D conformations using Open Babel. During this conversion, the 3D geometry of each molecule was optimized using the MMFF94 force field to ensure an accurate spatial configuration. The optimized 3D ligand structures were then processed using AutoDock Tools to generate the necessary input files for docking. The ligand molecules were imported into AutoDock Tools in PDB format, and all polar hydrogens were added to account for hydrogen bonding interactions. Torsion roots and rotatable bonds were detected automatically, with the number of torsions limited, if necessary, to prevent excessive flexibility. Gasteiger charges were computed and assigned to all ligand atoms, and non-polar hydrogens were merged to simplify the molecular representation without compromising interaction accuracy. The ligand molecules were then saved in PDBQT format. After preparation, each ligand PDBQT file was visually inspected in AutoDock Tools to confirm the accuracy of torsions, charges, and hydrogen placements. Any structural anomalies were corrected before proceeding to docking.

## Molecular docking

For blind docking, a grid box large enough to encompass the entire protein is defined to ensure the ligand can explore all potential binding sites. The grid size for the X, Y, and Z coordinates was set to 61, 75, and 71 Å, respectively, centralized at −31.65, 18.13, and 65.15. The grid spacing was set as 1.00 Å with the exhaustiveness of 8. The docking simulation was executed using AutoDock Vina [44], and the resulting binding poses were analyzed based on binding affinities and interaction profiles. We identified the key interacting residues and binding mode and exported the best-docked complex for further analysis in PyMOL or Discovery Studio.

## ADMET (Absorption, Distribution, Metabolism, Excretion, and Toxicity) properties

To evaluate the ADMET properties of a ligand molecule, we used online tools namely SwissADME (http://www.swissadme.ch/index.php) and pkCSM (https://biosig.lab.uq.edu.au/pkcsm/). By combining the results from both SwissADME and pkCSM, we gain a comprehensive understanding of the ADMET properties top hit, which is crucial for drug discovery and development.

## MD simulations

MD simulations of apo protein and ligand-protein complex were conducted using CHARMM36 force field in GROMACS, including CGenFF version 4.6 to generate ligand topology [45–48]. Each system was placed in a cubic box with a 10 Å distance to the edges before solvation with the TIP3P water model. Appropriate counter ions (Na+ and Cl−) were added to the systems to neutralize net charge, mimic physiological ionic strength, and improve simulation stability and accuracy. The energy minimization was performed using the steepest descent algorithm followed by conjugate gradient methods to eliminate unfavorable atom contacts. Systems were heated gradually from 0 K to 300 K and equilibrated for 100 ps at constant volume and pressure of 1-bar pressure to allow the solvent to reach equilibrium around the Ligand-protein complex. Finally, a 100 ns production run was performed at constant temperature and pressure.

## Analysis of MD simulations GROMACS data trajectories

The GROMACS trajectories were analysed to compute the Root Mean Square Deviation (RMSD), Root Mean Square Fluctuation (RMSF), Radius of gyration (Rg), and Solvent Accessible Surface Area (SASA), as well as perform hydrogen bond analysis and examine secondary structure changes in hIL1β when binding with a ligand molecule [49,50]. Principal Component Analysis was also conducted on the GROMACS trajectories to gain insights into the conformational dynamics and flexibility of biomolecules and identify the key motions that are relevant for their function. We employed the gmx anaeig command in GROMACS to perform various analyses on the eigenvalues and eigenvectors and calculating the free energy landscape.

## Molecular Mechanics Poisson–Boltzmann Surface Area (MM-PBSA) analysis

The MM-PBSA approach is a widely employed post-processing method used to estimate the binding free energy between interacting biomolecules [51]. MM-PBSA has gained popularity over the years because it balances computational efficiency and accuracy when evaluating biomolecular interactions, such as protein-ligand or protein-protein binding. In this study, MM-PBSA calculations were carried out using molecular dynamics trajectories, following the protocol described by Bhardwaj et al. [52]. The g_mmpbsa tool, an interface designed to integrate MM-PBSA calculations within the GROMACS molecular dynamics framework, was specifically used to estimate the binding free energies. The binding free energy ($\Delta G_{Binding}$) was calculated using the following equation: $\Delta G_{Binding}$ (total binding energy of the protein-ligand complex) = $G_{Complex} - [G_{Receptor}$ (binding energy of free receptor) + $G_{Ligand}$ (binding energy of the unbounded ligand)].

## Results

### Molecular docking and ADMET properties

The binding affinities of the top five ligand molecules screened against the hIL1β protein are summarized in Table 1. Among these, Quercetin demonstrated the highest binding affinity (−10.3 kcal/mol), significantly exceeding the other selected phytochemicals, namely Limonin, Narirutin, Chrysin, and Epicatechin. Notably, the binding energy of Quercetin was also more favorable than that of the positive control compound, T9C_168477827 (−9.5 kcal/mol), which was originally co-crystallized with hIL1β. This strong binding suggests a more stable and potentially effective interaction between Quercetin and the target site on hIL1β. The higher affinity may be attributed to Quercetin's polyphenolic structure, which allows for multiple hydrogen bond donors and acceptors, as well as π-π stacking and van der Waals interactions with key residues in the binding pocket. The results of the interaction patterns of top five hits are depicted in Fig 1A-F. We found that Quercetin formed several conventional hydrogen bonds with LYS94, MET95, PRO57, and SER45 amino acid residues of hIL1β (Fig 1E). LYS94 and MET95 lie in the C-terminal region of hIL1β, which is involved in receptor binding and structural integrity. These residues are typically highly conserved across species, as they participate in maintaining the correct folding and functional conformation required for cytokine-receptor interactions. PRO57 and SER45 are in the central β-strand regions, which are part of the core β-trefoil structure of hIL1β. Proline residues often serve critical structural roles due to their rigid ring structure, while serine residues can be sites for post-translational modifications like phosphorylation but are also frequently conserved when involved in hydrogen bonding or secondary structure stability. Based on sequence conservation, structural importance, and low mutation frequency in population data, LYS94, MET95, PRO57, and SER45 are likely conserved residues. Their involvement in stable hydrogen bonding with Quercetin suggests that the binding efficacy of quercetin is less likely to be affected by natural mutations. Therefore, Quercetin's interaction with these residues adds confidence to its potential as a robust therapeutic candidate with sustained binding efficacy, even across potential sequence variations. Additionally, several other bonds namely van der Waals, Pi Sigma, and Pi alkyl bonds were also formed between Quercetin and hIL1β. Given its favorable binding energy, structural complementarity with the target

**Table 1. Top five prioritized compounds based on their binding affinities towards hIL1β.**

| Name of the ligand | Binding Energy (kcal/mol) | pKi | Ligand Efficiency (kcal/mol/non-H atom) |
|---|---|---|---|
| Chrysin | −7.2 | 5.28 | 0.3789 |
| Epicatechin | −7.2 | 5.28 | 0.3429 |
| Limonin | −7.9 | 5.79 | 0.2324 |
| Narirutin | −7.7 | 5.65 | 0.1878 |
| Quercetin | −10.3 | 6.99 | 0.4182 |
| T9C_168477827 | −9.5 | 6.97 | 0.2714 |

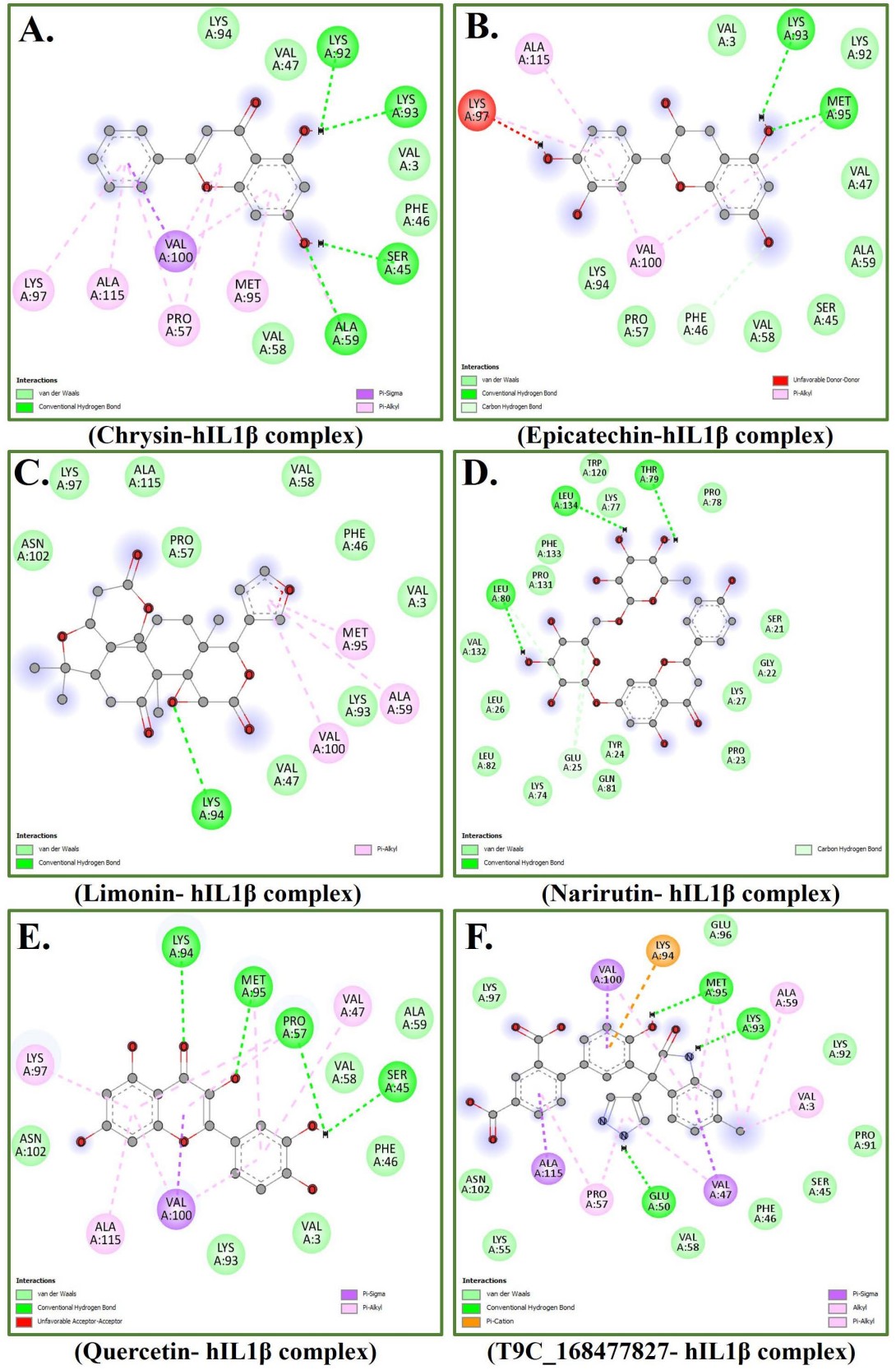

**(Chrysin-hIL1β complex)**

**(Epicatechin-hIL1β complex)**

**(Limonin- hIL1β complex)**

**(Narirutin- hIL1β complex)**

**(Quercetin- hIL1β complex)**

**(T9C_168477827- hIL1β complex)**

**Fig 1. (2D interactions) are illustrating bonding interactions. (A)** Chrysin-hIL1β complex. **(B)** Epicatechin hIL1β complex. **(C)** Limonin hIL1β complex. **(D)** Narirutin hIL1β complex. **(E)** Quercetin hIL1β complex. **(F)** T9C_168477827 hIL1β complex.

protein, and known anti-inflammatory properties, Quercetin was selected as the top hit for further in silico studies, to explore its potential as a candidate inhibitor of hIL1β-mediated inflammatory responses.

Results of ADMET properties of Quercetin are represented in Table 2. Results demonstrated favorable characteristics in terms of absorption, distribution, metabolism, excretion, and lack of toxicity. Water solubility was observed at −2.925 log mol/L, intestinal absorption at 77.207%, VDss at 1.559 log L/kg, BBB permeability at −1.098 log BB, total clearance at 0.407 log ml/min/kg, with no AMES toxicity or hepatotoxicity observed.

## MD simulations, RMSD, RMSF, Rg and SASA

To evaluate the structural stability and conformational behavior of the system, we analyzed the molecular dynamics (MD) simulation production data by calculating four key parameters: root mean square deviation (RMSD), root mean square fluctuation (RMSF), radius of gyration (Rg), and solvent-accessible surface area (SASA). The numerical values of these parameters are summarized in Table 3, while their temporal profiles are illustrated in Fig 2A-D. The average RMSD values were calculated to assess the overall structural deviation during the simulation. The apo hIL1β exhibited an average RMSD of 0.23 nm, whereas the hIL1β-Quercetin complex showed a lower average RMSD of 0.15 nm, suggesting that binding with Quercetin stabilized the protein structure and reduced its conformational drift over time. For the RMSF, which indicates the flexibility of individual residues, the average values were 0.11 nm for apo hIL1β and 0.09 nm for the hIL1β-Quercetin complex. The slightly higher RMSF in the complex suggests that Quercetin binding may have induced localized flexibility in certain regions of the protein. The Rg, a measure of the protein's compactness, remained consistent across both systems, with an average value of 1.51 nm, indicating that Quercetin binding did not significantly alter the overall structural compactness of hIL1β. Finally, the SASA, which reflects the extent of the protein surface exposed to the solvent, was slightly reduced in the Quercetin-bound form (91.29 nm2) compared to the unbound hIL1β (92.05 nm2).

**Table 2. In silico prediction of ADME & Tox properties for Quercetin (QT).**

| | Properties | | | | | | | | | | | | |
|---|---|---|---|---|---|---|---|---|---|---|---|---|---|
| | **Absorption** | **Distribution** | | | **Metabolism** | | | | | | | **Excretion** | **Toxicity** |
| **Models** | **Intestinal absorption (human)** | **VDss (human)** | **BBB permeability** | **CNS permeability** | **CYP** | | | | | | | **Total clearance** | **AMES toxicity/ Hepatoxicity** |
| | | | | | **Substrate** | | **Inhibitor** | | | | | | |
| | | | | | 2D6 | 3A4 | 1A2 | 2C19 | 2C9 | 2D6 | 3A4 | | |
| **Unity** | **Numeric (% absorbed)** | **Numeric (log L/kg)** | **Numeric (Log BB)** | **Numeric (Log PS)** | **Categorical (yes/no)** | | | | | | | **Numeric (log ml/min/ kg)** | **Categorical (yes/no)** |
| Predicted values | | | | | | | | | | | | | |
| QT | 77.207 | 1.559 | −1.098 | −3.065 | NO | No | Yes | No | No | No | Yes | 0.407 | No/No |

**Table 3. Parameters (Mean±SD) calculated for hIL1β and hIL1β-Quercetin complex from 100 ns MD simulations production run.**

| Systems | RMSD (nm) | RMSF (nm) | Rg (nm) | SASA (nm²) | Intramolecular H-bonds |
|---|---|---|---|---|---|
| hIL1β | 0.23±0.05 | 0.11±0.08 | 1.51±0.008 | 92.05±2.51 | 93 |
| hIL1β-Quercetin complex | 0.15±0.02 | 0.09±0.05 | 1.51±0.006 | 91.29±1.72 | 94 |

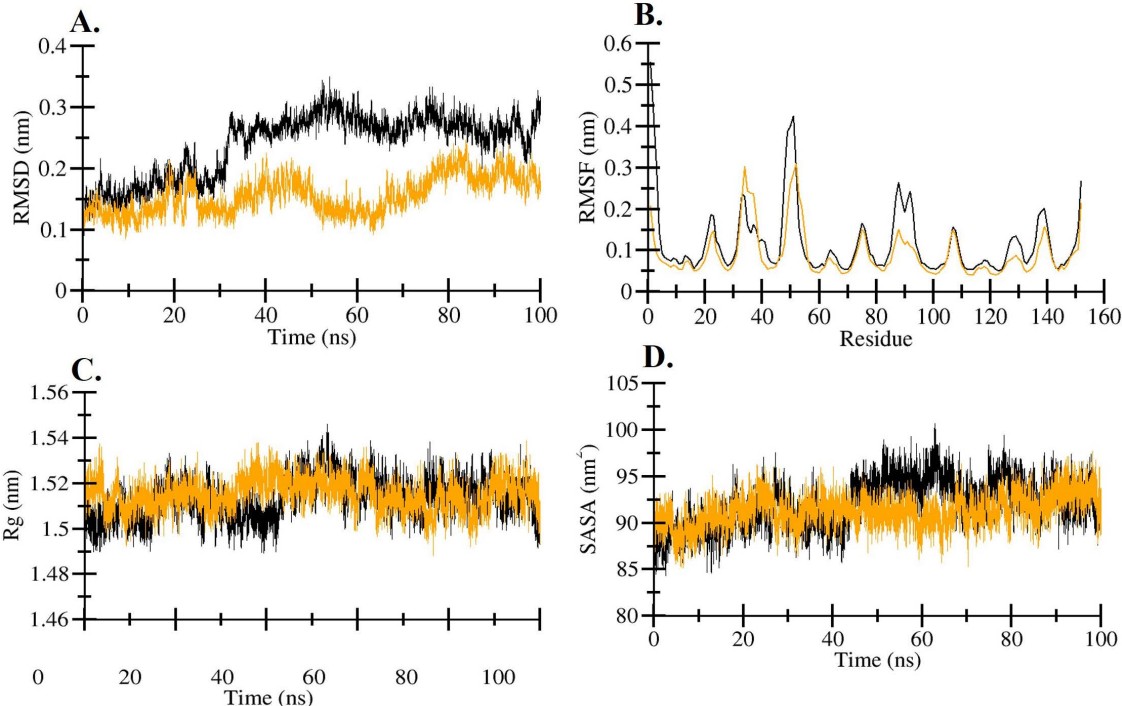

**Fig 2. Structural dynamics of hIL1β upon Quercetin binding as a function of time. (A)** RMSD plot of hIL1β and hIL1β-Quercetin complex. **(B)** RMSF plot of plot of hIL1β and hIL1β-Quercetin complex. **(C)** Rg plot of hIL1β and hIL1β-Quercetin complex. **(D)** SASA plot of hIL1β and hIL1β-Quercetin complex. Black and yellow colour represent hIL1β and hIL1β-Quercetin complex, respectively.

This decrease suggests a modest reduction in solvent exposure, possibly due to the burial of some surface residues upon quercetin binding.

### Intramolecular and intermolecular hydrogen bond analysis

The average number of intramolecular hydrogen bonds (H-bonds) within hIL1β was calculated to be 93 before binding with Quercetin and slightly increased to 94 after the binding event, as shown in Table 3 and illustrated in Fig 3A. This minor increase suggests that the interaction with Quercetin may contribute to a marginal stabilization of the hIL1β structure by enhancing internal hydrogen bonding. To further investigate the interaction between Quercetin and hIL1β, we analyzed the time-evolution of intermolecular hydrogen bonds throughout the molecular dynamics (MD) simulation. The results revealed that a single intermolecular H-bond was consistently maintained between Quercetin and hIL1β over the course of the simulation (Fig 3B), indicating a stable and persistent interaction between the ligand and the protein.

### Secondary structure analysis

Table 4 presents a summary of the average quantity of residues participating in the formation of various secondary structure elements, such as α-helices, β-sheets, and turns, throughout the simulation period. Fig 4A and 4B illustrate the structural organization of these key secondary elements in the native hIL1β protein and the hIL1β-Quercetin complex, respectively. The secondary structure profiles derived from the molecular dynamics simulation trajectories of both systems exhibited a stable and consistent pattern. This structural stability indicates that the essential secondary structure components of the protein remained well-preserved during the simulation, suggesting that binding of Quercetin did not induce significant conformational changes or destabilization in the overall secondary structure of hIL1β.

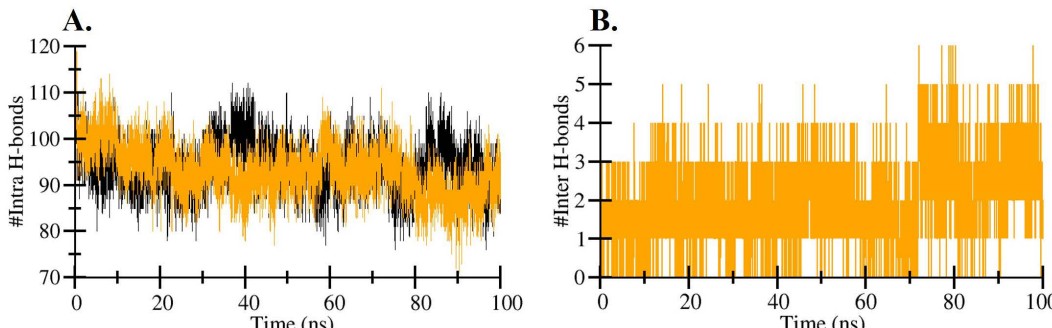

**Fig 3. Intra- and Intermolecular hydrogen bonds. (A)** Time evolution and stability of hydrogen bonds formed Intramolecular within hIL1β. **(B)** Time evolution and stability of hydrogen bonds formed intermolecular between hIL1β and hIL1β-Quercetin complex. Black, and yellow colour represent hIL1β and hIL1β-Quercetin complex, respectively.

**Table 4. The proportion of residues typically involved in forming secondary structure of the protein.**

| Percentage of protein secondary structure (SS %) | | | | | | | | | |
|---|---|---|---|---|---|---|---|---|---|
| **Systems** | **Structure** | **Coil** | **B -sheet** | **B-bridge** | **Bend** | **Turn** | **A-helix** | **3-helix** | **PPII-helix** |
| **hIL1β** | 0.63 | 0.18 | 0.47 | 0.01 | 0.14 | 0.13 | 0.02 | 0.02 | 0.03 |
| **hIL1β-Quercetin complex** | 0.63 | 0.18 | 0.48 | 0.02 | 0.13 | 0.13 | 0.01 | 0.04 | 0.02 |

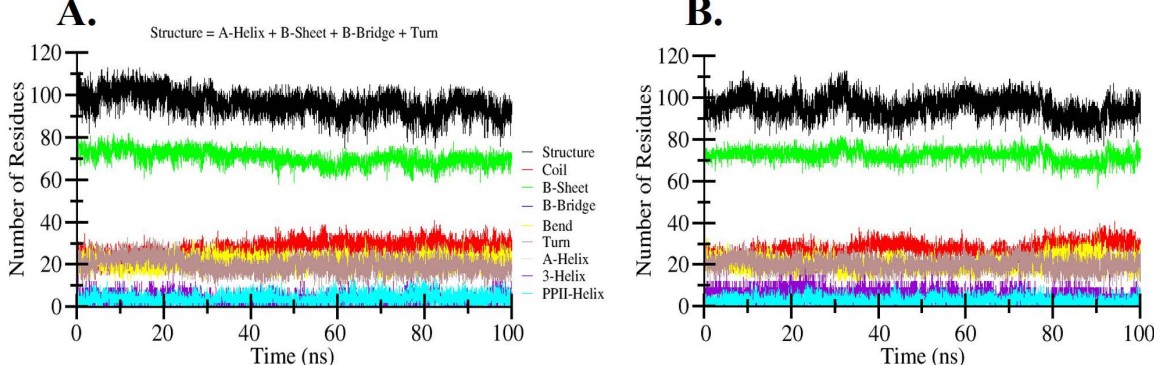

**Fig 4. Secondary structure plots indicating the structural elements present during the 100 ns MD simulations. (A)** hIL1β. **(B)** hIL1β-Quercetin complex.

## Principal component analysis

We have investigated the conformational sampling and general movements of hIL1β and hIL1β-Quercetin complex using the fundamental dynamics technique. Fig 5A & 5B depict the conformational sampling of the hIL1β and hIL1β-Quercetin complex in the subspace. In addition to two different EVs projected by the protein Cα atoms, the projection implies the conformational sampling of hIL1β. In this instance, we found that the hIL1β-Quercetin complex occupied a notably different conformation within decreased subspace compared to the unbound hIL1β. As seen in Fig 5A, the phase space of free hIL1β and stable clusters significantly overlapped, reducing the overall flexibility of the hIL1β-Quercetin complex at

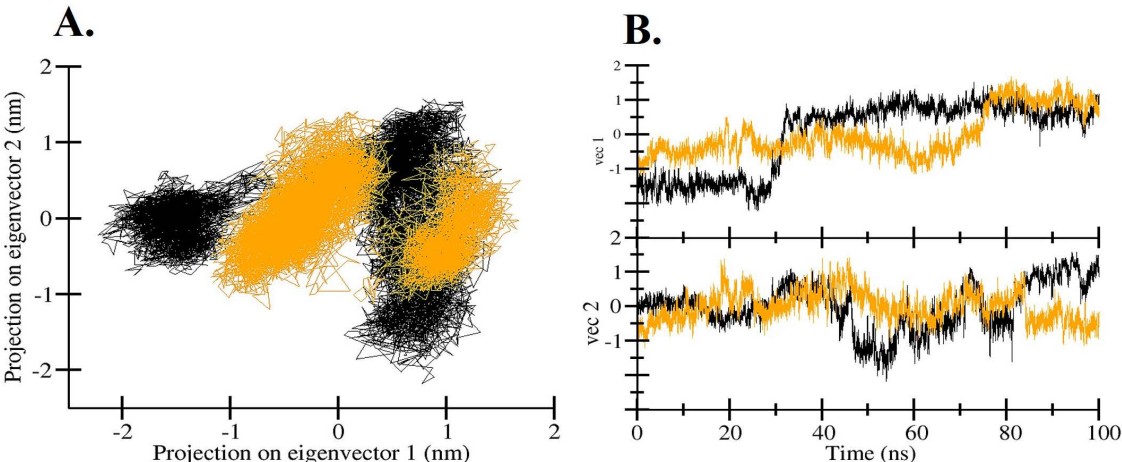

**Fig 5. Principal Component Analysis. (A)** 2D projections of trajectories on eigenvectors showed different projections of hIL1β. **(B)** The projections of trajectories on eigenvectors with respect to time. Black and yellow colour represent hIL1β and hIL1β-Quercetin complex, respectively.

PC1. Therefore, the PCA analysis indicates that the structural arrangement of hIL1β in conjunction with quercetin is rather stable and has reduced dynamics.

### Gibbs free energy landscapes (FELs)

To better examine the conformational dynamics and stability of hIL1β before and after quercetin binding, the Gibbs free energy landscapes (FELs) were created and assessed using the first two EVs. The FELs of hIL1β and hIL1β-Quercetin complex are displayed in Fig 6A-D. The more intense blue colour in graphs indicates different conformational states with lower energy. The size and location of the phases contained inside a single stable global minimum are marginally changed by quercetin's binding with hIL1β, according to FEL plots. Fig 6A & 6C clearly show that hIL1β has a single global minimum contained within a single basin. Comparably, when hIL1β binds to quercetin, it displays several states but retains a single global minimum with two to three local basins with different populations (Fig 6B & 6D). Overall, the analysis of FELs shows that quercetin's binding to hIL1β does not result in protein unfolding in the simulation.

### MM-PBSA analysis

We also evaluated the binding free energy associated with the interaction between human hIL1β and Quercetin using the MM-PBSA (Molecular Mechanics Poisson–Boltzmann Surface Area) method. The results, expressed as mean ± standard deviation (SD), are summarized in Table 5. The analysis revealed that all components of the binding energy namely van der Waals, electrostatic, polar solvation, and non-polar solvation energies contributed favorably to the overall interaction. This suggests a stable and energetically favorable binding between hIL1β and quercetin.

### Discussion

The development of new therapeutics targeting IL-1β is critical for managing post-treatment endodontic diseases due to the significant role this cytokine plays in inflammation and bone resorption [17]. Elevated IL-1β levels contribute to persistent inflammation and delayed healing in periapical tissues following endodontic treatment [7]. Targeting IL-1β can help modulate the inflammatory response, reduce tissue destruction, and promote regeneration [53]. Innovative approaches, such as IL-1β inhibitors or antagonists, can provide more effective treatment options, potentially leading to better clinical outcomes and reduced recurrence rates. Molecular docking and MD simulations plays a crucial role in drug design and

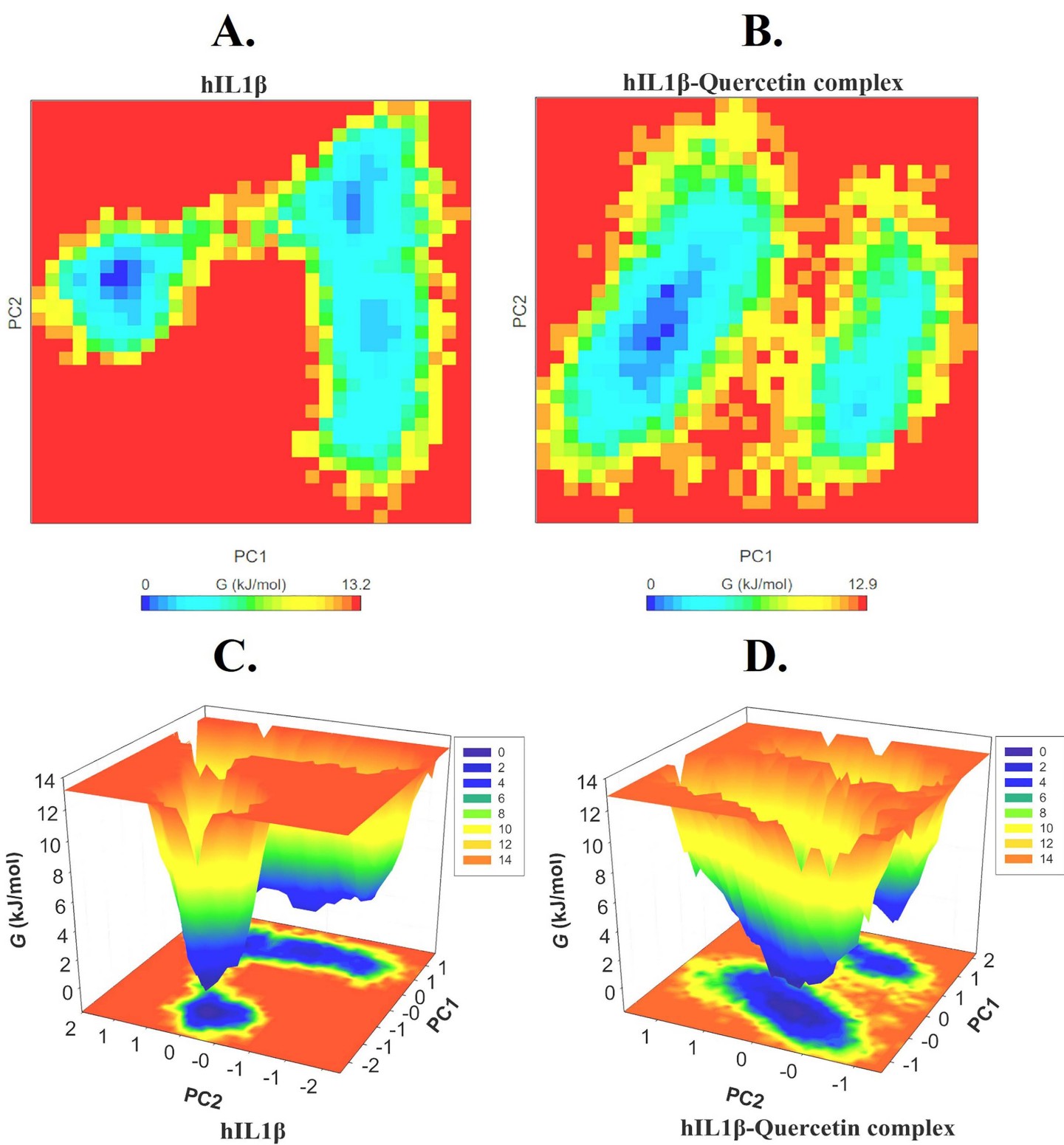

**Fig 6. Gibbs energy landscape plots obtained during 100 ns MD simulations.** (A and C) hIL1β. (B and D) hIL1β-Quercetin complex.

**Table 5. Binding free energy (Mean±SD) calculations for hIL1β-Quercetin complex using MM-PBSA calculations.**

| Complex | ΔE_Binding (kJ/mol) | SASA (kJ/mol) | ΔE_Polar solvation (kJ/mol) | ΔE_Electrostatic (kJ/mol) | ΔE_Van der Waal (kJ/mol) |
|---|---|---|---|---|---|
| hIL1β-quercetin complex | −199.426±13. 312 | −17.664±2.121 | 128.856±18.385 | −48.757±12.356 | −255.215±21.513 |

development by predicting the optimal binding orientation of drug candidates to their target proteins [49,50]. This helps in identifying potent inhibitors, optimizing lead compounds, and understanding the molecular basis of drug-receptor interactions, thus accelerating the drug discovery process. Unlike prior studies that primarily focused on downstream cytokine suppression and canonical pathway inhibition [29,31], our work employs molecular docking, dynamics simulations, and other in silico-based techniques to characterize the direct interaction between quercetin and IL-1β at the molecular level. This allows us to propose a direct inhibitory binding mechanism, which extends existing findings based on functional outcomes alone. Most existing studies have examined quercetin's effects on IL-1β-induced signaling, rather than quercetin as a direct inhibitor of IL-1β itself. Our study provides evidence for quercetin's direct binding affinity toward IL-1β, thereby offering a mechanistic perspective that may help guide future structure-based drug design efforts.

The molecular docking results indicated that Quercetin exhibited the highest binding affinity to hIL1β among the top five hits, surpassing Limonin, Narirutin, Chrysin, and Epicatechin. Furthermore, Quercetin's binding affinity exceeded that of the positive control molecule, T9C_168477827. The interaction analysis revealed that Quercetin formed several conventional hydrogen bonds with key amino acid residues of hIL1β, namely LYS94, MET95, PRO57, and SER45. Additionally, van der Waals, Pi Sigma, and Pi alkyl bonds contributed to the stability of the Quercetin-hIL1β complex. These observations underline Quercetin's strong binding capability and potential therapeutic relevance in targeting hIL1β. In addition to this, review of literature reflects that this flavonoid molecule "Quercetin" has significant therapeutic potential due to its wide range of biological activities. For example, it possesses strong antioxidant (reviewed in [54]), anti-inflammatory (reviewed in [55]), and anticancer properties, making it valuable in the prevention and treatment of various diseases. Quercetin exhibited favorable ADMET properties (Table 2), demonstrating promising characteristics in terms of absorption, distribution, metabolism, excretion, and lack of toxicity. The comprehensive interaction of Quercetin with hIL1β, significant notions from the review literature, and ADMET properties justifies the selection of Quercetin and hIL1β for MD simulation to explore their dynamic behavior and potential therapeutic relevance.

MD simulations of apo proteins and ligand-protein complexes play a pivotal role in drug design and development by offering insights into the dynamic interactions between drugs and their target proteins [56]. MD simulations allow for the exploration of protein flexibility, ligand binding modes, and the stability of ligand-receptor interactions over time [57]. Analysis of data trajectories of MD simulation production helps identify key binding sites, optimize drug candidates and aids in developing therapies. MD simulation data provided valuable insights into the structural stability and dynamics of the hIL1β-Quercetin complex. RMSD measures the overall deviation of the ligand-protein complex from its initial structure over time [49]. A stable RMSD suggests that the complex has reached an equilibrium, while large fluctuations may indicate instability or conformational changes [58]. Consistent low RMSD values generally reflect a stable binding of the ligand to the protein [59]. The RMSD value was found comparatively lower in case of hIL1β-Quercetin complex, suggesting that the ligand is effectively binding and stabilizing the protein in a specific conformation [59]. This reduced RMSD indicates that the ligand has induced a conformational change in the protein, which is typically characteristic of antagonistic action. The antagonist ligand likely prevents the protein from adopting its active conformation, thereby inhibiting its function. RMSF measures the flexibility of individual residues in the protein structure during the simulation [60]. High RMSF values at certain residues suggest regions of the protein that are more flexible or undergo large conformational changes, which could be critical for binding interactions or function [61]. Lower RMSF values indicate stable, rigid regions of the protein [62]. Although the average RMSF value for the complex was slightly higher than that for apo hIL1β, this suggests some flexibility in the complex, which may facilitate binding interactions. Slightly higher RMSF values are consistent

with an antagonist action where the ligand prevents full activation or induces partial disruptions in the protein's structure, making it flexible in regions necessary for its biological activity. Rg gives insight into the compactness or expansion of the protein-ligand complex. A stable or decreasing Rg indicates that the protein-ligand complex maintains a compact structure, which is often a sign of stability [63]. An increasing Rg may suggest unfolding or structural relaxation, which could impact the binding of the ligand [64]. The Rg value remained same in both of the cases, apo hIL1β and hIL1β-Quercetin complex. If the Rg of the ligand-protein complex is similar to that of the apo protein in the context of antagonist action, it suggests that the ligand binding does not affect the overall compactness or conformation of the protein, reflecting minimal structural conformational change and non-disruptive binding. SASA measures the surface area of the protein or ligand that is exposed to solvent [65]. The average SASA for hIL1β-Quercetin complex was found to slightly decreased compared to apo hIL1β. The SASA value for hIL1β-Quercetin complex was comparatively lower apo hIL1β. A decrease in SASA upon ligand binding typically suggests that the ligand interacts deeply with the protein, burying some of the protein's surface area [66].

The intramolecular and intermolecular hydrogen bond analysis provides valuable information on the structural integrity, binding affinity, and interaction dynamics of the ligand-protein complex, crucial for rational drug design and development [67,68]. If the number of these bonds remains constant or increases upon ligand binding, it suggests that the protein maintains its structural integrity in the complex [69]. Hydrogen bond analysis revealed an increase in the number of intramolecular hydrogen bonds within hIL1β upon Quercetin binding, from 93 to 94. This increase may contribute to the stabilization of the protein structure. The consistent presence of a single intermolecular hydrogen bond between Quercetin and hIL1β throughout the MD simulations highlights the stability of the binding interaction.

By analyzing the secondary structure of the apo protein, one can assess its inherent stability and flexibility [70,71]. Key secondary structure elements (alpha-helices, beta-sheets, turns, and random coils) should ideally remain stable over the simulation time if the protein is inherently stable [49]. When a ligand binds to the protein, it may stabilize or destabilize certain regions [50]. Comparing the secondary structure of the ligand-bound protein to the apo protein can reveal these changes. A stabilizing ligand might reduce fluctuations in certain secondary structures, while a destabilizing ligand might induce more structural changes [72,73]. The secondary structure analysis indicated that the secondary structure elements of hIL1β were conserved throughout the simulation, both in the free state and when bound to Quercetin. This suggests that Quercetin binding does not significantly alter the secondary structure of hIL1β, maintaining its structural integrity. The secondary structure elements produced by the simulated trajectories for both systems showed a consistent pattern, reflecting that protein secondary structure elements were conserved throughout the simulation.

PCA can be used to ascertain the overall aggregate motions of the C atoms in a protein, as indicated by the EVs of the covariance matrix [74]. It is used to investigate the general kinetics and conformational sampling of proteins as well as those of protein-ligand complexes [75,76]. PCA has been shown to be an excellent technique for analyzing how proteins fold in presence of small ligand molecules [77]. PCA demonstrated that the hIL1β-Quercetin complex occupied a different conformation within a decreased subspace compared to the unbound hIL1β. The reduced overall flexibility of the hIL1β-Quercetin complex, as evidenced by the overlapping phase space and stable clusters, suggests a more stable structural arrangement upon Quercetin binding. Therefore, the PCA analysis indicates that the structural arrangement of hIL1β in conjunction with quercetin is rather stable and has reduced dynamics. To better examine the conformational dynamics and stability of hIL1β before and after quercetin binding, the Gibbs free energy landscapes (FELs) were created and assessed using the first two EVs. FEL analysis further confirmed the stability and conformational dynamics of the hIL1β-Quercetin complex. The FEL plots indicated that hIL1β, both in its free state and when bound to Quercetin, retained a single global minimum, reflecting a stable conformation. The presence of several local basins in the hIL1β-Quercetin complex suggests that Quercetin binding induces minor conformational changes without causing protein unfolding.

This study, while providing valuable computational insights into the interaction between quercetin and IL-1β, has some limitations. Firstly, a detailed structure-activity relationship (SAR) analysis of the selected flavonoids was not performed,

limiting a deeper understanding of the chemical features influencing activity. Secondly, the molecular dynamics simulations were restricted to a 100 ns timeframe, which may not fully capture the conformational stability of the protein-ligand complex, as suggested by the observed RMSD fluctuations. Additionally, the specificity of quercetin binding to IL-1β was not assessed in comparison to other cytokines or potential off-target proteins, and no experimental validation, such as in vitro or in vivo assays, was conducted to corroborate the in-silico predictions. The study also does not rule out the possibility of compensatory biological mechanisms or resistance developing against quercetin-based therapy upon chronic exposure. Moreover, pharmacokinetic challenges such as low bioavailability and rapid metabolism of Quercetin remain unaddressed experimentally, raising questions about its clinical applicability. While the study highlights potential therapeutic relevance, further research involving experimental validation, off-target screening, pharmacokinetic profiling, and delivery strategy optimization is essential to translate these findings into clinical settings.

## Conclusion

In conclusion, this study highlights IL1β as a critical mediator in endodontic inflammation and a promising therapeutic target. Through molecular docking and molecular dynamics simulations, quercetin was identified as the top candidate among 329 plant-derived compounds, exhibiting the highest binding affinity and stable interaction with hIL1β. MD simulation parameters confirmed the structural stability of the quercetin-hIL1β complex, with minimal conformational changes and consistent hydrogen bonding. These findings support the potential of quercetin as a stable and effective small-molecule inhibitor of IL1β, offering a promising lead for the development of novel therapeutic strategies for endodontic diseases and PTED.

## Acknowledgments

The authors are thankful to Deanship of Graduate Studies & Scientific Research, Jazan University, Saudi Arabia for providing the facility.

## Author contributions

**Conceptualization:** nezar Boreak, Shroog Ali Almasoudi, Abdulelah Alharbi.

**Formal analysis:** Shahd Tahrei, Atyaf Abu eishah.

**Funding acquisition:** nezar Boreak, Taghreed Ahmed Madkhali, Mashael Ali Hattan.

**Investigation:** nezar Boreak, Shroog Ali Almasoudi, Maryam Hassan Majrashi.

**Methodology:** Huda Ali Daak, Amani Hakami.

**Project administration:** nezar Boreak, Shroog Ali Almasoudi.

**Visualization:** Mona Judayba, Shahd Tahrei, Atyaf Abu eishah.

**Writing – original draft:** nezar Boreak.

**Writing – review & editing:** Abdulelah Alharbi, Mona Judayba, Shahd Tahrei, Atyaf Abu eishah.

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
