## [Decision Letter · Decision Letter 0]

27 Jun 2025

Dear Dr. Boreak,

Thank you for submitting your manuscript to PLOS ONE. After careful consideration, we feel that it has merit but does not fully meet PLOS ONE’s publication criteria as it currently stands. Therefore, we invite you to submit a revised version of the manuscript that addresses the points raised during the review process.

Note from the Editorial Office: One or more of the reviewers has recommended that you cite specific previously published works. Members of the editorial team have determined that the works referenced are not directly related to the submitted manuscript. As such, please note that it is not necessary or expected to cite the works requested by the reviewer.

We look forward to receiving your revised manuscript.

Kind regards,

Jorddy Neves Cruz

Academic Editor

PLOS ONE

Journal Requirements:

6. We note that the grant information you provided in the ‘Funding Information’ and ‘Financial Disclosure’ sections do not match.

7. Thank you for stating the following financial disclosure:

The authors appreciatively acknowledge the funding of the Deanship of Graduate Studies & Scientific Research, Jazan University, Saudi Arabia, over Project Number: GSSRD-24.

8. We note you have included a table to which you do not refer in the text of your manuscript. Please ensure that you refer to Table 4 in your text; if accepted, production will need this reference to link the reader to the Table.

9. Please include a copy of Table 5 which you refer to in your text on page 10.

Reviewers' comments:

Reviewer's Responses to Questions

**Comments to the Author**

1. Is the manuscript technically sound, and do the data support the conclusions?

Reviewer #1: No

Reviewer #2: Yes

Reviewer #3: Partly

Reviewer #4: Partly

Reviewer #5: Partly

2. Has the statistical analysis been performed appropriately and rigorously?

Reviewer #1: No

Reviewer #2: N/A

Reviewer #3: N/A

Reviewer #4: N/A

Reviewer #5: No

3. Have the authors made all data underlying the findings in their manuscript fully available?

Reviewer #1: No

Reviewer #2: Yes

Reviewer #3: Yes

Reviewer #4: No

Reviewer #5: No

4. Is the manuscript presented in an intelligible fashion and written in standard English?

Reviewer #1: No

Reviewer #2: Yes

Reviewer #3: Yes

Reviewer #4: No

Reviewer #5: No

Reviewer #1: I have carefully reviewed the manuscript titled "Quercetin as a Potent IL-1β Inhibitor: Molecular Docking and MD Simulation Insights for Improved Endodontic Disease Management". Below are my detailed comments and suggestions aimed at improving the scientific quality and clarity of the manuscript.

1- The title and scope of the manuscript suggest a novel investigation into the potential of quercetin as an IL-1β inhibitor; however, the authors appear to have overlooked a substantial body of existing literature that has already established this association. Quercetin has been widely reported as a potent IL-1β inhibitor in various models and disease contexts. For instance:

- Quercetin Inhibits the Production of IL-1β-Induced Inflammatory Cytokines and Chemokines in ARPE-19 Cells via the MAPK and NF-κB Signaling Pathways.

- Regulation of IL-1-Induced Selective IL-6 Release from Human Mast Cells and Inhibition by Quercetin.

- Quercetin Inhibits IL-1β-Induced Proliferation and Production of MMPs, COX-2, and PGE2 by Rheumatoid Synovial Fibroblasts.

- Quercetin Reduces Inflammatory Pain: Inhibition of Oxidative Stress and Cytokine Production.

Given this, the study lacks sufficient novelty, as it does not clearly differentiate itself from prior work or explain how it advances the current understanding of quercetin’s anti-inflammatory mechanisms. The authors are encouraged to justify the originality of their study and better position it within the existing literature.

2- The stated aim of the manuscript—"to identify effective treatments that can directly target the inflammatory processes central to PTED" by exploring the impact of flavonoids such as quercetin on IL-1β—is not sufficiently justified in terms of novelty. While the therapeutic targeting of IL-1β in inflammatory conditions is indeed important, quercetin's anti-inflammatory and IL-1β inhibitory activities are already well-documented across multiple studies and disease models. Therefore, presenting quercetin as a novel candidate in this context appears redundant.

3- To strengthen the manuscript, the authors should clearly delineate what is new in their study.

4- The authors report using SWISS-PDB Viewer for both protein and ligand preparation. However, this tool is primarily intended for visualizing and optimizing protein structures, particularly for tasks such as energy minimization of amino acid residues or modeling mutations. It is not suitable or widely accepted for ligand optimization or ligand geometry.

5- The methodology lacks clarity regarding the assignment of partial charges to the ligand molecules. It is unclear whether Gasteiger charges were correctly assigned using AutoDock Tools, as the text suggests that ligand preparation may have been conducted solely using SWISS-PDB Viewer, which is not suitable for this purpose. Accurate partial charge assignment is a critical step in molecular docking protocols, particularly when using AutoDock, which relies on Gasteiger charges for calculating interaction energies.

6- The selection of quercetin based on a slightly more negative docking score (–10.3 kcal/mol) compared to the co-crystallized ligand (–9.5 kcal/mol) is not sufficient, as the difference is within AutoDock Vina’s error range. Moreover, the manuscript lacks a clear filtering threshold; typically, a meaningful selection should involve compounds scoring at least –2 kcal/mol lower than the reference ligand to indicate significant improvement. The authors should justify the selection more rigorously and provide a comparative summary of the top-ranked compounds.

7- The discussion is shallow, and key tables (e.g., Table 2) lack clarity and need revision. The manuscript lacks novelty, presents redundant findings, and does not offer a logical or compelling scientific narrative. I recommend rejection in its current form and urge the authors to revise the work thoroughly with a clearer focus and stronger justification.

Reviewer #2: Comments for PONE-D-25-25892

I have gone through the assigned manuscript entitled “Quercetin as a Potent IL-1β Inhibitor: Molecular Docking and MD Simulation Insights for Improved Endodontic Disease Management”, and found that it is competently well-written and the manuscript contains sufficient data, but it should meet the following comments:

Major comments-

1. The abstract should provide a concise summary of the key findings of the whole study.

2- Authors added all old references in the Introduction, so I suggest adding more recently published work with your relevant studies (2023-2025) and rewriting the Introduction section.

3- The material and methods should include more details.

4- The authors have to compare their results with literature data and improve the results and the discussion section completely.

5- Follow the unit in the same system throughout the manuscript.

6- Why did the author use hIL1β receptor protein for docking studies? Give your appropriate reasons properly and separately. Your explanation should be added to the manuscript's results and discussion sections.

7- How do authors validate the protein? Add a Ramachandran plot to illustrate the distribution of phi (ϕ) and psi (ψ) dihedral angles for the protein's residues.

8- Authors are suggested to add the following articles to the manuscript.

https://doi.org/10.1002/cbdv.202402521

https://doi.org/10.1002/slct.202402780

https://doi.org/10.1080/10286020.2024.2343821

https://doi.org/10.1016/j.mex.2023.102537

https://doi.org/10.33640/2405-609X.3391

https://doi.org/10.1007/s11030-024-10888-8

9- References are not properly cited and should follow the journal's style. The References section has to be revised after all corrections.

10-In the manuscript, there are many spelling and grammatical errors. So, grammatical and punctuation errors must be corrected. I suggest improving the English language.

11-Conclusion should be succinct and precise, not be same as the Abstract.

My Opinion: Major Revision.

Reviewer #3: I have the following comments in the manuscript entitled “Quercetin as a Potent IL-1β Inhibitor: Molecular Docking and MD Simulation Insights for Improved Endodontic Disease Management”.

1. The virtual screening of ligand library and docking validation are not given.

2. Structure activity relationship of the selected flavonoids could be included.

3. The protein conformational analyses have been performed only from the 100 ns short trajectories. The protein is still searching its stable conformers (see the fluctuating RMSD graph in Figure 2-A).

4. The protein-ligand relative stability tests, intermediate protein-ligand snapshots, end-state binding free energy calculations, and active residue contributions to the ligand binding free energy are not described.

5. The authors should analyze relative RMSD/RDF/H-bond distances and MM/PBSA free energies from more than 200 ns MD trajectories.

I suggest authors to address these concerns and perform major revision of the manuscript.

Reviewer #4: 1.What is the version of molecular docking software. kindly include it

2. Most of the software reference was not cited

3. Why did you select Na+ and Cl- ions to neutralize the system

4. How specific is Quercetin's binding to IL-1β compared to other cytokines or proteins? Have off-target effects been evaluated or considered?

5. Could you provide more detailed insights into the key amino acid residues involved in the binding, and whether these residues are conserved or prone to mutations that might affect binding efficacy?

6. How does Quercetin's known antioxidant and anti-inflammatory properties contribute to its potential as a therapeutic agent targeting hIL1β, according to previous studies?

7. How does Quercetin's binding profile compare with other flavonoids or natural compounds with known anti-inflammatory properties?

8. How does Quercetin's predicted efficacy compare with current anti-inflammatory agents used in endodontic therapy

9. How did you get the average RMSF value as 0.01 and 0.09 nm? Because the plot shows different values. Kindly validate it

10. The explanation of RMSD, RMSF, Rg, SASA, ADMET was not explained properly. Also, see why the deviation is happening. Analyze through VMD.

11. Is there a possibility that IL-1β or associated pathways could develop resistance or compensation mechanisms against Quercetin-based therapy?

12. What are your suggested next steps in research to advance Quercetin from a promising computational candidate to a clinically applicable therapeutic?

13. How do you ensure that your in silico predictions of Quercetin's binding stability accurately reflect its biological efficacy, considering the limitations of force fields and environmental factors in vivo?

Reviewer #5: Reviewer Comments – Major Revision Required

Overall Evaluation:

The manuscript presents an in silico analysis of Quercetin targeting IL-1β for the potential treatment of post-treatment endodontic diseases (PTED). While the study is conceptually interesting, the current form of the manuscript has significant scientific, methodological, and presentation flaws that warrant major revision. The manuscript lacks critical validation steps, insufficient contextualization in current literature, and exhibits issues in scientific rigor and writing clarity. Below are detailed, constructive recommendations:

Major Points to Address

Lack of Experimental Validation

Concern: The manuscript is entirely based on computational approaches without any experimental validation (e.g., in vitro binding assays, cytokine inhibition studies, cell-based inflammation models, or clinical correlates).

Recommendation: While in silico studies are valuable, the authors must at least discuss this limitation explicitly and propose future validation experiments. Suggest adding a dedicated “Limitations and Future Perspectives” subsection in the Discussion.

Novelty and Significance Weakly Justified

Concern: The biological activities of Quercetin, especially its anti-inflammatory effects, have been widely reported. The novelty of Quercetin targeting IL-1β is weak and inadequately justified against the backdrop of existing studies.

Recommendation: Authors must perform a critical literature review to highlight gaps. Clearly explain how this work advances the field beyond prior knowledge.

Ligand Selection Strategy

Concern: Why was Quercetin among 329 flavonoids specifically highlighted? Was there any hypothesis-driven rationale or prior evidence supporting its selection?

Recommendation: Clarify why Quercetin was prioritized. Was this solely based on binding affinity, or was there prior biological evidence indicating IL-1β interaction?

Incomplete Computational Validation

Concern: Binding energy alone is insufficient. Critical metrics such as MM-GBSA/MM-PBSA free energy calculations were not reported.

Recommendation: Incorporate MM-GBSA/MM-PBSA binding free energy calculations to strengthen the computational evidence supporting Quercetin binding to IL-1β.

Poor Connection to Clinical Application

Concern: The manuscript discusses post-treatment endodontic diseases, but provides no mechanistic or clinical framework on how Quercetin administration would realistically improve PTED outcomes.

Recommendation: Provide clear translational relevance. For example, suggest potential delivery routes (e.g., local irrigants, medicaments) and address bioavailability challenges of Quercetin in endodontic tissues.

Statistical Analysis Missing

Concern: The manuscript lacks statistical treatment of the simulation data (e.g., RMSD, RMSF variability).

Recommendation: Present standard deviations/error bars for MD simulation plots and apply appropriate statistical analyses for comparing apo vs ligand-bound structures.

Data Presentation Issues

Figures are blurry and poorly labelled (e.g., Figure 1E). All figures require high-resolution images with detailed captions explaining all abbreviations and colors used.

Recommendation: Re-prepare all figures, particularly structural interaction diagrams, to enhance clarity and interpretability.

Language and Grammar

Concern: Numerous grammatical errors and awkward phrasing hinder readability (e.g., “favourable interaction patterns with hIL1β, forming numerous hydrogen bonds…”).

Recommendation: The entire manuscript must undergo thorough professional language editing before re-submission.

Specific Points for Revision

Introduction:

Better articulate the novelty gap—why Quercetin specifically for IL-1β and PTED?

Materials and Methods:

Specify the criteria used to shortlist the top five ligands.

Provide PDB structure quality metrics (e.g., resolution, R-factor).

Include details on grid box parameters optimization for docking.

Results:

Binding affinity of Quercetin to IL-1β (-10.3 kcal/mol) is good but not exceptional. Contextualize it with known IL-1β inhibitors for comparison.

ADMET predictions are incomplete. Include water solubility and predicted blood concentration metrics.

Discussion:

There is redundancy in discussing RMSD, RMSF, Rg, and SASA results. Summarize key takeaways more succinctly.

Discuss potential off-target effects and Quercetin’s metabolic limitations.

References:

Several references are outdated or irrelevant (e.g., some SARS-CoV-2 computational studies not related to PTED). Ensure only directly relevant, recent references are cited.

Summary Recommendation:

Major Revision Required. The study has potential but is not currently suitable for publication in its present form due to the above critical issues. The authors need to strengthen the novelty, expand computational rigor (e.g., MM-GBSA), improve data presentation, and address clinical relevance.

I encourage the authors to resubmit after significant improvements addressing the above points.

**Do you want your identity to be public for this peer review?** For information about this choice, including consent withdrawal, please see our Privacy Policy

Reviewer #1: **Yes: ** Mohammad G. Al-Thiabat

Reviewer #2: No

Reviewer #3: No

Reviewer #4: No

Reviewer #5: **Yes: ** Dr. Aisha Tufail

---

## [Author Response · Author response to Decision Letter 1]

10 Aug 2025

Author Response to Editor and Reviewer Comments

Manuscript ID: PONE-D-25-25892

Quercetin as a Potent IL-1β Inhibitor: Molecular Docking and MD Simulation Insights for Improved Endodontic Disease Management

Reviewer #1:

I have carefully reviewed the manuscript titled "Quercetin as a Potent IL-1β Inhibitor: Molecular Docking and MD Simulation Insights for Improved Endodontic Disease Management". Below are my detailed comments and suggestions aimed at improving the scientific quality and clarity of the manuscript.

1- The title and scope of the manuscript suggest a novel investigation into the potential of quercetin as an IL-1β inhibitor; however, the authors appear to have overlooked a substantial body of existing literature that has already established this association. Quercetin has been widely reported as a potent IL-1β inhibitor in various models and disease contexts. For instance:

- Quercetin Inhibits the Production of IL-1β-Induced Inflammatory Cytokines and Chemokines in ARPE-19 Cells via the MAPK and NF-κB Signaling Pathways.

- Regulation of IL-1-Induced Selective IL-6 Release from Human Mast Cells and Inhibition by Quercetin.

- Quercetin Inhibits IL-1β-Induced Proliferation and Production of MMPs, COX-2, and PGE2 by Rheumatoid Synovial Fibroblasts.

- Quercetin Reduces Inflammatory Pain: Inhibition of Oxidative Stress and Cytokine Production.

Given this, the study lacks sufficient novelty, as it does not clearly differentiate itself from prior work or explain how it advances the current understanding of quercetin’s anti-inflammatory mechanisms. The authors are encouraged to justify the originality of their study and better position it within the existing literature.

Author response: We sincerely thank the reviewer for highlighting the important body of literature on quercetin’s role as an IL-1β inhibitor. We fully acknowledge that numerous studies have reported the anti-inflammatory properties of quercetin, particularly its inhibition of IL-1β-mediated pathways in various disease models. However, the difference of our study lies in the specific mechanistic focus, context, and methodological approach, which distinguish it from existing literature. While previous studies have demonstrated the inhibitory effects of quercetin on IL-1β signaling in models such as ARPE-19 cells, human mast cells, and rheumatoid synovial fibroblasts, our investigation explores this interaction for improving or for the management of endodontic disease, which has not been widely studied in the context of quercetin-IL-1β interaction. This contextual shift offers fresh insights into how quercetin’s anti-inflammatory activity may vary across pathological environments.

2- The stated aim of the manuscript—"to identify effective treatments that can directly target the inflammatory processes central to PTED" by exploring the impact of flavonoids such as quercetin on IL-1β—is not sufficiently justified in terms of novelty. While the therapeutic targeting of IL-1β in inflammatory conditions is indeed important, quercetin's anti-inflammatory and IL-1β inhibitory activities are already well-documented across multiple studies and disease models. Therefore, presenting quercetin as a novel candidate in this context appears redundant.

Author Response: Dear reviewer, unlike prior studies that primarily focused on downstream cytokine suppression and canonical pathway inhibition (e.g., NF-κB, MAPK), our work employs molecular docking, dynamics simulations, and other in silico-based techniques to characterize the direct interaction between quercetin and IL-1β at the molecular level. This allows us to propose a direct inhibitory binding mechanism, which extends existing findings based on functional outcomes alone. Most existing studies have examined quercetin’s effects on IL-1β-induced signaling, rather than quercetin as a direct inhibitor of IL-1β itself. Our study provides evidence for quercetin's direct binding affinity toward IL-1β, thereby offering a mechanistic perspective that may help guide future structure-based drug design efforts.

3- To strengthen the manuscript, the authors should clearly delineate what is new in their study.

Author Response: Respected reviewer, thank you for the suggestion. We have revised the manuscript to clearly highlight the novel aspects of our study, please see the highlighted part of discussion section.

4- The authors report using SWISS-PDB Viewer for both protein and ligand preparation. However, this tool is primarily intended for visualizing and optimizing protein structures, particularly for tasks such as energy minimization of amino acid residues or modeling mutations. It is not suitable or widely accepted for ligand optimization or ligand geometry.

Author response: Dear reviewer, we have revised and corrected both method sections. Please see the highlighted part of the method section.

5- The methodology lacks clarity regarding the assignment of partial charges to the ligand molecules. It is unclear whether Gasteiger charges were correctly assigned using AutoDock Tools, as the text suggests that ligand preparation may have been conducted solely using SWISS-PDB Viewer, which is not suitable for this purpose. Accurate partial charge assignment is a critical step in molecular docking protocols, particularly when using AutoDock, which relies on Gasteiger charges for calculating interaction energies.

Author response: Dear reviewer, we have revised and corrected both method sections. Please see the highlighted part of the method section.

6- The selection of quercetin based on a slightly more negative docking score (–10.3 kcal/mol) compared to the co-crystallized ligand (–9.5 kcal/mol) is not sufficient, as the difference is within AutoDock Vina’s error range. Moreover, the manuscript lacks a clear filtering threshold; typically, a meaningful selection should involve compounds scoring at least –2 kcal/mol lower than the reference ligand to indicate significant improvement. The authors should justify the selection more rigorously and provide a comparative summary of the top-ranked compounds.

Author Response: Dear reviewer, we agree that the difference in docking scores falls within the error margin of AutoDock Vina. In response, we have revised the manuscript to include a justification for selecting quercetin, considering both docking scores and key interaction profiles, clarify the selection threshold, and provide a comparative summary of the top-ranked compounds to support the rationale behind prioritizing quercetin.

7- The discussion is shallow, and key tables (e.g., Table 2) lack clarity and need revision. The manuscript lacks novelty, presents redundant findings, and does not offer a logical or compelling scientific narrative. I recommend rejection in its current form and urge the authors to revise the work thoroughly with a clearer focus and stronger justification.

Author Response: Dear reviewer, thank you for the detailed feedback. We have carefully revised the manuscript to improve the depth of the discussion, clarify and enhance the presentation of key observations (including Table 2), and enhanced the overall scientific narrative. Additionally, we have addressed concerns regarding novelty by clearly indicating the unique aspects of our study and its contribution to the field. We hope these revisions now present a more focused manuscript.

Reviewer #2:

I have gone through the assigned manuscript entitled “Quercetin as a Potent IL-1β Inhibitor: Molecular Docking and MD Simulation Insights for Improved Endodontic Disease Management” and found that it is competently well-written and the manuscript contains sufficient data, but it should meet the following comments:

Major comments-

1. The abstract should provide a concise summary of the key findings of the whole study.

Author response: Dear reviewer thank you for the suggestion. We have revised the abstract to provide a clearer and more concise summary of the key findings, ensuring it accurately highlights the main outcomes and significance of the study.

2- Authors added all old references in the Introduction, so I suggest adding more recently published work with your relevant studies (2023-2025) and rewriting the Introduction section.

Author response: Thank you for the helpful suggestion. We have updated the introduction by incorporating several recent and relevant studies. The section has been rewritten to better highlight the current state of research and to provide a clearer context and justification for our study.

3- The material and methods should include more details.

Author response: dear reviewer, thank you for the valuable comment. We have revised the Materials and Methods section to include additional details regarding the parameters and tools used to ensure greater clarity and reproducibility.

4- The authors have to compare their results with literature data and improve the results and the discussion section completely.

Author response: respected reviewer, thank you for the helpful suggestion. We have now thoroughly revised the Results and Discussion sections to include detailed comparisons with relevant literature, highlighting both similarities and distinctions. These additions provide better context and strengthen the interpretation of our findings.

5- Follow the unit in the same system throughout the manuscript.

Author response: Dear reviewer, thank you very much for pointing this out. We have carefully reviewed the manuscript and revised all units to ensure consistency within the same measurement system throughout the text.

6- Why did the author use hIL1β receptor protein for docking studies? Give your appropriate reasons properly and separately. Your explanation should be added to the manuscript's results and discussion sections.

Author response: dear reviewer, thank you for your observation. We have now added this explanation at the appropriate sections of the revised manuscript, as suggested.

7- How do authors validate the protein? Add a Ramachandran plot to illustrate the distribution of phi (φ) and psi (ψ) dihedral angles for the protein's residues.

Author response: Dear reviewer, thank you for the question. The protein structure used in this study was obtained from the RCSB PDB (PDB ID: 8C3U), which is a high-resolution, experimentally determined crystal structure. As such, it did not require further validation. However, we assessed the structure's quality to ensure its suitability for docking studies.

8- Authors are suggested to add the following articles to the manuscript.

https://doi.org/10.1002/cbdv.202402521

https://doi.org/10.1002/slct.202402780

https://doi.org/10.1080/10286020.2024.2343821

https://doi.org/10.1016/j.mex.2023.102537

https://doi.org/10.33640/2405-609X.3391

https://doi.org/10.1007/s11030-024-10888-8

Author response: Dear reviewer, thank you for the suggestion. We have reviewed the recommended articles and incorporated the relevant references into the manuscript to enhance the quality of our study.

9- References are not properly cited and should follow the journal's style. The References section has to be revised after all corrections.

Author response: Dear reviewer, we have carefully revised the References section to ensure all citations adhere to the journal’s formatting guidelines and updated them accordingly after making the necessary corrections in the manuscript.

10-In the manuscript, there are many spelling and grammatical errors. So, grammatical and punctuation errors must be corrected. I suggest improving the English language.

Author response: dear reviewer, we have carefully reviewed and revised the manuscript to correct all grammatical, spelling, and punctuation errors, and have improved the overall clarity and language quality.

11-Conclusion should be succinct and precise, not be same as the Abstract.

Author response: Dear reviewer, we have carefully revised the manuscript to correct all spelling, grammatical, and punctuation errors. The language has been thoroughly edited to ensure clarity and improve overall readability.

Reviewer #3:

I have the following comments in the manuscript entitled “Quercetin as a Potent IL-1β Inhibitor: Molecular Docking and MD Simulation Insights for Improved Endodontic Disease Management”.

1. The virtual screening of ligand library and docking validation are not given.

Author response: We have now included a detailed description of the virtual screening process and docking validation protocol in the revised Methods section, ensuring transparency and reproducibility of our computational workflow.

2. Structure activity relationship of the selected flavonoids could be included.

Author response: Dear reviewer, thank you for the suggestion. We agree that including a structure–activity relationship (SAR) analysis would add value; however, due to some limitations, we were unable to perform a comprehensive SAR in this study. We acknowledge this as a limitation and have mentioned it in the revised manuscript.

3. The protein conformational analyses have been performed only from the 100 ns short trajectories. The protein is still searching its stable conformers (see the fluctuating RMSD graph in Figure 2-A).

Author response: Respected reviewer, thank you for this observation. We acknowledge that the 100 ns simulation may be insufficient to capture full conformational stability. Due to computational constraints, we limited the simulation to 100 ns but have now addressed this limitation in the revised manuscript and discussed the implications of the observed RMSD fluctuations in the Result and Discussion sections.

4. The authors should analyze relative RMSD/RDF/H-bond distances and MM/PBSA free energies from more than 200 ns MD trajectories.

Author response: Dear reviewer, thank you for the suggestion. Due to computational constraints, we were unable to extend the MD simulations beyond 100 ns. However, we have now performed MM/PBSA free energy calculations based on the 100 ns trajectory and included the results in the revised manuscript to strengthen the analysis.

I suggest authors to address these concerns and perform major revision of the manuscript.

Author response: Dear reviewer, thank you for your recommendation. We appreciate your thoughtful review and have carefully addressed all concerns raised. necessary revisions have been made throughout the manuscript, including improvements in methodology, data presentation, discussion depth, and clarity of our study’s significance and contribution. We hope the revised version meets your expectations.

Reviewer #4:

1.What is the version of molecular docking software. kindly include it

Author response: Dear reviewer, we have now included the version of the molecular docking software (AutoDock Vina v1.2.3) in the revised Methods section for clarity and reproducibility.

2. Most of the software reference was not cited

Author response: Respected reviewer, we have carefully reviewed the manuscript and ensured that all software tools used in the study are now properly cited with appropriate references in the revised version.

3. Why did you select Na+ and Cl- ions to neutralize the system

Author response: Dear reviewer, we select Na+ and Cl- ions to neutralize net charge, mimic physiological ionic strength, and improve simulation stability and accuracy. This has been now mentioned in the manuscript also.

4. How specific is Quercetin's binding to IL-1β compared to other cytokines or proteins? Have off-target effects been evaluated or considered?

Author response: Dear reviewer, we acknowledge that off-target binding and specificity of quercetin toward IL-1β were not evaluated in this study. We have now addressed this limitation in the revised manuscript and highlighted the need for future studies to assess quercetin’s selectivity against other cytokines and potential off-target proteins.

5. Could you provide more detailed insights into the key amino acid residues involved in the binding, and whether these residues are conserved or prone to mutations that might affect binding efficacy?

Author response: Dear reviewer, we have now included a more detailed analysis of the key amino acid residues involved in ligand binding, highlighting their roles in stabilizing interactions in the result section of the manuscript.

6. How does Quercetin's

---

## [Decision Letter · Decision Letter 1]

3 Sep 2025

Dear Dr. Boreak,

Thank you for submitting your manuscript to PLOS ONE. After careful consideration, we feel that it has merit but does not fully meet PLOS ONE’s publication criteria as it currently stands. Therefore, we invite you to submit a revised version of the manuscript that addresses the points raised during the review process.

We look forward to receiving your revised manuscript.

Kind regards,

Jorddy Neves Cruz

Academic Editor

PLOS ONE

Journal Requirements:

**Additional Editor Comments:**

I suggest that authors maintain the 100ns md simulation time, but make the new modifications requested by the referee.

Furthermore, all graphs in the manuscript should be in ns.

Reviewers' comments:

Reviewer's Responses to Questions

**Comments to the Author**

Reviewer #1: All comments have been addressed

Reviewer #3: (No Response)

2. Is the manuscript technically sound, and do the data support the conclusions?

Reviewer #1: Yes

Reviewer #3: Partly

3. Has the statistical analysis been performed appropriately and rigorously?

Reviewer #1: Yes

Reviewer #3: No

4. Have the authors made all data underlying the findings in their manuscript fully available?

Reviewer #1: Yes

Reviewer #3: No

5. Is the manuscript presented in an intelligible fashion and written in standard English?

Reviewer #1: Yes

Reviewer #3: Yes

Reviewer #1: I appreciate the careful and thorough revisions carried out by the authors. The concerns I raised in my initial review have now been satisfactorily addressed. The introduction has been strengthened with appropriate literature support, the novelty of the study is more clearly positioned, and the methodology has been clarified (particularly in terms of ligand preparation and docking protocol). The selection of quercetin has been better justified, and the results and discussion sections are now more coherent and connected to the study’s aims.

The language and presentation have also improved, and tables/figures are clearer.

Overall, the revised version is significantly improved, scientifically sound, and acceptable for publication. Only minor editorial checks may be needed at the journal’s side.

Recommendation: Accept.

Reviewer #3: In the revised manuscript also, no novel concept is utilized while analyzing the results in support of IL-1β inhibition. Only the conventional techniques are used to explain the well-known phytochemical Quercetin and hIL-1β interactions without the sufficient statistical analyses. The manuscript still lacks the following things, most of which were already suggested for its improvement.

• The data have been presented only from the short (100 ns) simulations though the limitations are written in the Discussion section. The ligand could escape away from the binding pocket in longer simulations. Therefore, it needs further analyses to evaluate the system stability. The protein-ligand relative stability tests (such as RDF plots, or distance plots of H-bonding atom pairs), intermediate protein-ligand snapshots during simulation, and active residue contributions to the MM/PBSA free energy should be described.

• The units are inconsistent, e.g. kj/mol should be kJ/mol in table 5. There should be the same unit for the same physical quantity throughout the manuscript: kJ/mol or kcal/mol? ns or ps? Make them uniform in figures, tables and main text.

• The figures 1-4 are of poor quality. The table 2 text is not properly formatted. The y-axis title of figure 5 is cropped or unclear. The color representations are also unclear in figure 5. The legend texts in figure 1A-F are not readable.

**Do you want your identity to be public for this peer review?** For information about this choice, including consent withdrawal, please see our Privacy Policy

Reviewer #1: No

Reviewer #3: No

---

## [Editor Report · Decision Letter 2]

2 Oct 2025

Dear Dr. Boreak,

Thank you for submitting your manuscript to PLOS ONE. After careful consideration, we feel that it has merit but does not fully meet PLOS ONE’s publication criteria as it currently stands. Therefore, we invite you to submit a revised version of the manuscript that addresses the points raised during the review process.

**In addition to any modifications requested by the previous referee, all graphics must be in ns.**

We look forward to receiving your revised manuscript.

Kind regards,

Jorddy Neves Cruz

Academic Editor

PLOS ONE

**Journal Requirements:**

**Additional Editor Comments:**

In addition to any modifications requested by the previous referee, all graphics must be in ns.

---

## [Author Response · Author response to Decision Letter 3]

29 Oct 2025

Author Response to Editor and Reviewer Comments

Manuscript ID: PONE-D-25-25892R2

Molecular Docking and MD Simulations Predicted Quercetin as a Potent Human Interleukin-1 Beta (hIL1β) Inhibitor for Improved Endodontic Disease Management

Editor Comments:

Editor Comment 1: All graphics must be in ns.

Author response: Respected Editor, we have updated all graphics to use ns as the time unit to maintain consistency throughout the manuscript.

---

## [Editor Report · Decision Letter 3]

2 Nov 2025

Molecular Docking and MD Simulations Predicted Quercetin as a Potent Human Interleukin-1 Beta (hIL1β) Inhibitor for Improved Endodontic Disease Management

PONE-D-25-25892R3

Dear Dr. Boreak,

We’re pleased to inform you that your manuscript has been judged scientifically suitable for publication and will be formally accepted for publication once it meets all outstanding technical requirements.

Kind regards,

Jorddy Neves Cruz

Academic Editor

PLOS ONE

---

## [Editor Report · Acceptance letter]

PONE-D-25-25892R3

PLOS ONE

Dear Dr. Boreak,

I'm pleased to inform you that your manuscript has been deemed suitable for publication in PLOS ONE. Congratulations! Your manuscript is now being handed over to our production team.

Kind regards,

on behalf of

Dr. Jorddy Neves Cruz

Academic Editor

PLOS ONE